# Snaking into the Gothic: Serpentine Sensuousness in Lewis and Coleridge

Jeremy Chow 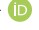

English Department, Bucknell University, Lewisburg, PA 17837, USA; j.chow@bucknell.edu

**Abstract:** This essay charts the ways late-eighteenth-century Gothic authors repurpose natural histories of snakes to explore how reptile-human encounters are harbingers of queer formations of gender, sexuality, and empire. By looking to M.G. Lewis's novel *The* Monk (1796) and his understudied short story "The Anaconda" (1808), as well as S.T. Coleridge's *Christabel* (1797–1800), I centre the last five years of the eighteenth century to apprehend the interwoven nature of Gothic prose, poetry, and popular natural histories as they pertain to reptile knowledge and representations. Whereas Lewis's short story positions the orientalised anaconda to upheave notions of empire, gender, and romance, his novel invokes the snake to signal the effusion of graphic eroticisms. Coleridge, in turn, invokes the snake-human interspecies connection to imagine female, homoerotic possibilities and foreclosures. Plaiting eighteenth-century animal studies, queer studies, and Gothic studies, this essay offers a *queer eco-Gothic* reading of the violating, erotic powers of snakes in their placement alongside human interlocutors. I thus recalibrate eighteenth-century animal studies to focus not on warm-blooded mammals, but on cold-blooded reptiles and the erotic effusions they afford within the Gothic imaginary that repeatedly conjures them, as I show, with queer interspecies effects.

**Keywords:** Gothic; queer; M.G. Lewis; Samuel Taylor Coleridge; reptile; snake; sexuality

"We are reptiles, miserable sinful creatures." —Father Jerome, *The Castle of Otranto*

## 1. Introduction

In *A Dictionary of Natural History* (1785), Scottish priest and bibliophile William Fordyce Mavor defines the anaconda as a "Ceylonese serpent of enormous magnitude, extremely mischievous among cattle" (Mavor 1785, p. 27). Also known as the "anacandaia" and the "bubalinus serpens", Mavor's explication curiously reiterates a second-hand anecdote to situate ethological knowledge about this particular East Indian serpent. Perhaps reflected by his incredulity, Mavor cannot corroborate the anecdotal evidence, which was "said to be written by an English gentleman resident in the East Indies, and signed R. Edwin, [and] was inserted in the *Edinburgh Evening Courant* of 15 August 1768" (p. 27). News of this particular anaconda vignette, as I show here, was not only long-lived, but was disseminated and published widely throughout Great Britain, so much so that it became a narrative *tour de force*, insinuating itself in the Gothic literary landscape on which I focus. Stationed as a colonial convoy in Ceylon (what is now Sri Lanka), the narrator—assumedly Edwin—is captivated by an animate branch that "bend[s] and twist[s]" against the forces of nature and weather, up a massive palm tree. A Ceylonese attendant asks Edwin what has so captured his attention, and in pointing out the strange sight, "a paleness overspread his whole face, and he seemed almost sinking to the earth with terror" (p. 27). The branch is no branch at all; it is a massive anaconda. The attendant demands that the house be shut up and all entries barred. Edwin balks, and thus witnesses the serpent snatch a small animal and retreat with it up the palm. A group of twelve (the Christian numerical allegory should not be lost on readers) assemble to "destroy it" (p. 27). They prove to be poor shots and save the task for another day, but not before Edwin embarks on a nearly 250-word long description of the anaconda's beauty—"It's [sic] back was [more] beautiful than can

well be imagined"—which is realised by its polychromatic scales that swirl with green, yellow streaks, black spots, and "dusky olive", black, and "great round long blotches of a perfect blood colour" (p. 27). The following day, Edwin observes the anaconda stalk and prey upon a tiger, graphically suffocating it and snapping its bones. Fattened and lethargic from the meal, the anaconda rests, and the Ceylonese snatch the opportunity to bash the anaconda and feast upon it. "It's [sic] length was thirty-three feet four inches", Edwin, *The Edinburgh Evening Courant*, and Mavor's *New Dictionary of Natural History*, in turn, report.

Even more curious, Matthew Gregory Lewis's understudied, Gothic short story, "The Anaconda: An East Indian Tale" (1808) reimagines much of Edwin's observations of the anaconda in a way that could be described as intertextuality, at best, and plagiarism, at worst. A framed narrative, very much like *The Monk* (1796) to which I will return later, the protagonist Everard Brooke returns from Ceylon with incredible wealth under suspicious circumstances. He is accused of homicide and to vindicate his name, he provides a history of his Ceylon travels entitled, "The Narrative of Everard Brooke". Like Edwin in *The Edinburgh Evening Courant*, Everard notices a "large excrescence" and a Ceylonese attendant, Zadi, "pronounced with difficulty—'The Anaconda! That is the Anaconda!—We are undone!'" (Lewis 1845, pp. 12, 13). Zadi's colonial "master" is held hostage by the serpent who looms in the favoured palm trees. The anaconda ensconces herself with the intent to kill. Zadi reveals:

> "The Anaconda was still employed in twisting itself in a thousand coils among the palm-branches with such restless activity, with rapidity so inconceivable, that it was frequently impossible for the sight to follow her movements. At *one* moment, she fastened herself by the end of her tail to the very summit of the loftiest tree, and stretched out at her whole length, swung backwards like the pendulum of a clock, so that her head almost seemed to graze the earth beneath her: then in *another* moment, before the eye was aware of her intention, she totally disappeared among the leafy canopies." (p. 17; italics original)

In her intractable and unpredictable mobilities, Lewis embraces the anaconda—using she/her pronouns to describe the snake, a fine point to which I will turn later—as the sublime idol. Everard narrates, "It was a sight calculated to excite in equal degrees of our horror and sour admiration: it united the most singular and brilliant beauty with every thing, that could impress the beholder with apprehension; and though while gazing upon it I felt that every limb shuddered involuntarily, I was still compelled to own, that never had I witnessed an exhibition more fascinating or more gratifying to the eye" (p. 17). Based on this description, it's difficult not to see Lewis's anaconda channelling Edmund Burke's (1757) assessment of power and its sublime corollaries. In his discussion of terror, Burke writes, "There are many animals, who though far from being large, are yet capable of raising ideas of the sublime, because they are considered as objects of terror. As serpents and poisonous animals of almost all kinds" (Burke 2008, p. 53). The serpent's sublimity lies in its poisonous affinities and hair-raising snake-human encounters induce worlds of terror. Everard's reiteration of sublime trembling imbues the anaconda with a pleasurable and horrific potency that lures and abjects the human-reptile connection.

"The Anaconda" fictionalises the witness-based natural history (whose veracity we may also question) on which Lewis's short story pends, and imbues the Ceylonese 'monster' with calculated sentience, vengeance, an insatiable hunger for human and animal bodies (she kills two men and swallows a dog and a bull), and in one comically preposterous moment, "her eyes, blazing with their own vindictive fires, shot lightnings through the gloom of night" (p. 24). Lewis's Gothic imagination deifies the anaconda, who, of course dies, and like Edwin's retelling, becomes a feast for the Ceylonese.[1] As Robert James

---

[1] The short story's prominence (and indeed Mavor's natural history) is reflected in Richard Milliken's companion melodrama, *Anaconda, the Terrific Serpent of Ceylon* first published in 1825 but which was performed at the West-London Theatre in October 1822 as reviewed by *The Lady's Magazine; or Entertaining Companion for the Fair Sex* (1822). The play seems to have given birth to an effective, mechanized serpent: 'The monster [the snake] is well represented by machinery, by which it is enabled to make a furious spring' (p. 574).

Merrett (1992) noted, natural history and the eighteenth-century novel are two widely popularised narrative forms that do not run parallel; they are, instead, intersecting—or coiled together, to continue with the snake puns. Whereas Merrett emphasises the class-based ideologies and reading publics that engage and collapse with both genres, I weave a different story here, namely, to document the repeated slithering of serpentine representations in Gothic literatures that realise queer potentialities. To be clear, I am not interested in outing the snake, examining it exclusively as a phallic symbol, or reading the snake's death as indicative of castration anxiety, which would seemingly rehash long-held Freudian associations that manifest in "Medusa's Head" (1922). Snakes are priapic figures; we need not be reminded of the lyrics to Sir Mix-A-Lot's "Baby Got Back" (1992). These things we know.

This essay realises that a queer animal studies approach is germane to the progression of the Gothic throughout the late eighteenth and early nineteenth centuries, but also to fear-inducing creepy crawlies, like snakes, that can participate in growing interspecies intimacies within Gothic studies.[2] Demonic, feminised, mythic, and violating snakes appear widely in proto-Gothic and Gothic narratives, worming their way through John Milton's *Paradise Lost* (1667), with Gustave Doré's engravings (1866) to boot, through S.T. Coleridge's *Rime of the Ancient Mariner* (1798), also engraved by Doré in 1884, from John Keats's Lamia (1820) to P.B. Shelley's "The Assassins" (1814), *Laon and Cynth* (1817), and *On the Medusa of Leonardo di Vinci* (1824).[3] To snake into the Gothic is not singularly to trace the recurrence of slithering beings throughout Gothic formations; "snaking in" demands that we reckon with the uncomfortable sensuous semiotics that accompany these recurrent representations, which often coincide with the snake's alleged penchant for bodily violation and the induction of trauma. "Snaking into the Gothic" proposes a means by which we magnify queer forms of human and nonhuman being that are performed by literature's repeated attention to concatenations of reptile and human bodies.

The Gothic snakes that I trace here—epitomised by Lewis's "The Anaconda", Lewis's *The Monk*, and Coleridge's *Christabel* (1797–1800)—demonstrate the mutually informative ways by which the explosion of late-eighteenth-century natural history influenced poetic and fictional depictions that further located animal lore within the popular cultural imaginary. The animal turn within eighteenth- and nineteenth-century studies has long cultivated an attraction to warm-blooded mammals, and the sheer amount of work on pet-keeping practices is evidence of the enrapturing charisma of domesticates, which is rivalled only by scholarship on histories of the great apes, to which I have also unabashedly and admittedly contributed (Festa 2019; Tague 2015; Brown 2010; Keenleyside 2016; Cole 2016; Palmeri 2006; Nash 2003; Chow 2021). I document how natural histories account for the snake, especially the anaconda, and seek to grow the seeds of influence planted by Hall (2018), Hobbins (2017), Pacyga (2015), and Chakrabarti (2012) to ascertain a more capacious grasp of how reptilian imagery, in general, and snake representations, in particular, inform notions of gender, sexuality, and empire in the late-eighteenth and early-nineteenth centuries.

"Snaking into the Gothic" is a model for a *queer eco-Gothic*—a hybrid that melds the eco-Gothic with the queer Gothic. By Lisa Kröger's (2013) assessment, an eighteenth-century eco-Gothic may appear in the architectural ruins reclaimed by nature, whereas the eco-Gothic tropes I underline here account for the ways in which nonhuman and human life jointly participate in intimacies that do not default to squabbles over triumph. I am not interested, in other words, in visualising a melee for human or reptile supremacy as much as unravelling the queer effects of that melee. In this vein, I take up and extend a process of queering the Gothic that has received continued and varied attention (Haggerty 2006;

---

2    I follow the lead of Mel Chen (2012); Jennifer Terry (2000); Myra Hird (2006) and Alice Kuzniar (2008) who deftly consider how human-animal eroticisms may inform the triangulation of race and affect, fields of science, trans theories and embodiments, and plural affections with domesticates, respectively.

3    See Teddi Lynn Chichester (1996) for an overview of Shelley's repeated conjuring of snakes, which, for Chichester, becomes imbued with a masculine ethos. See Jerome McGann (1972) for a seminal recuperation of the Romantic Medusa and her reverberations through Pater, Swinburne, and Goethe.

Hughes and Smith 2009; Fincher 2007; Haefele-Thomas 2012; Zigarovich 2017). William Hughes and Smith (2009) describe queering the Gothic as recognition of how "the queer may be said to effectively deconstruct the very standards by which its own 'deviance' is reckoned and quantified. If the queer state may persist successfully, even if only for a short disruptive period, then it retains the potential to construct itself as a visible alternative to all that is not-queer" (p. 4). This essay looks to those short disruptive periods in which snakes rear their troubling heads. In specific, I underline a queer multispecies Gothic approach that closely examines the substitutive heteronormative work and its concomitant queer resonances that emerge from "The Anaconda", the graphically erotic invocations of the serpent in *The Monk*, and I close with the homoerotic and female intimacies of multispecies affections in *Christabel*. If, as George Haggerty (2006) proffers, the queer Gothic rejects the codification of sexual norms as well as a hetero/homo binary—it lives, instead, in desire— then "Snaking into the Gothic" unveils the modes of desirousness that surface when we unveil reptile and human entanglements. By uniting queer and eco-Gothic lines of inquiry, this essay bespeaks a commitment to reimagining textures and terrains of eroticism that invariably enfold human, nonhuman, and animal(ised) bodies, so as to further decentre a hetero/homo binary and envisage spectrums of intimate inter-relationality that are not bound by species. Put another way, if queering the eighteenth century intends to explode finite boundaries of gendered and sexual containment, then interspecies eroticisms dissolve the loose reins of human supremacy that too often plague queer and environmental studies.

## 2. Natural Hiss-tory

Despite Edwin's and Lewis's adamance to position the anaconda deep within the orientialised fecundity of Ceylon, you will not find anacondas there. Anacondas are exclusively autochthonous to South America. In other words, what Edwin and Everard experience climbing and rappelling from palm trees is not an anaconda. However, the word "anaconda", which now exclusively refers to a massive snake species in South America and brings to mind a series of shoddy, demonising Hollywood films by the same name, is a direct descendant of Edwin's narrative, at least as corroborated by the *Oxford English Dictionary*. As fin-de-siecle herpetologist Frank Wall writes in *Ophidia Taprobanica*; or, *the Snakes of Ceylon*, anaconda is a portmanteau of the Tamil words for 'elephant' (*anai*) and 'killer' (*kolra/kondra*) (Wall 1921, p. 48). The *Oxford English Dictionary* reveals that because of the false cultural and linguistic etymologies, the word "anaconda" became a substitute for "any large snake that crushes its prey" (Anaconda 2020). The reptile that Edwin and Everard bear witness to is the python—a species that can share similar colouration, sizes, and penchants for constricting its prey. Edwin and Everard encounter a python in misattributed anaconda's clothing.

While Hall (2018) has noted that late-eighteenth-century Britain's engagement with large and venomous snakes was overwhelmingly focused on the Indian subcontinent (as a result of the expansion of the East Indian Company), in the last years of the eighteenth century, the widespread fears of snakes and anacondas, in particular, become visually ingrained in the imperial cultural imaginary as a result of, I contend, John Gabriel Stedman's *Narrative of a Five Years' Expedition Against the Revolted Negroes of Surinam* (1796)—set not in the East Indies, but the West Indies. Completed in 1790 and published in full in the same year in which Lewis's *The Monk* was released, 1796, Stedman's opus reveals a graphic display of a captured and hanged anaconda—truly an anaconda this time—that reverberated through the broader European and transatlantic worlds. As Diana Donald (2007) observes, in the emergence of eighteenth-century natural history (beginning with the century itself) through the prominence the genre acquired by the end of the century, "visual imagery was not simply dependent on, or subsidiary to, a written text: in itself it constituted an important means of advancing knowledge" (p. 29). Stedman's *Narrative* alongside William Blake's accompanying engravings collectively promote two intercon-nected yet separate forms of literacy that offer manifold apprehensions of the imperial human-reptile connection.

Blake's engraving, "The Skinning of the Aboma Snake, shot by Cpt. Stedman," shown in Figure 1, locates the anaconda within its natural habitat, vexed by colonial forces.[4] Traversing the Coermoetibo Creek to patrol for marooned slaves, and inflamed with a fever, Stedman narrates the appearance of an anaconda—or aboma, the term preferred by the Surinamese—that is mistaken, because of its colour, for a black, enslaved body. David, one of Stedman's attendant slaves, disabuses the febrile emissary that the mirage is not an escaped slave, it is instead an amphibious snake. Stedman's fear of missing out (FOMO) demands that he ignore his illness to shoot the aboma for sport; unfortunately, he proves repeatedly throughout his voyages to be a bad marksman. This characterisation further insinuates itself in Lewis's "The Anaconda", wherein Everard similarly acknowledges his sharpshooting failures: "I was an excellent marksman, and was certain, that I had pointed my piece exactly at the monster's head: and ye . . . whether too great anxiety made my hand shake, or that the animal at that very moment made some slight change in her attitude, I know not" (p. 18). The anaconda proves, in doubled fashion, to beset colonial concentration and induce anxieties among imperial cynosures. For Stedman, it takes four total bullets, but only David is the truly successful marksman; in other words, the black slave has a better grasp of colonial commodities—a point I take up in another essay on Stedman's racial invocations and imperial calculus.[5] Stedman, David, and the other slaves noose and drag the aboma (still alive) alongside the boat before arriving in Barbacoeba, where the snake is skinned, its internal organs eviscerated and mined for oil (a sort of medical treatment), and its flesh divvied up to be consumed—the reptile measuring "22 feet and some inches, and its thickness like that of my black Boy Quacoo, who might then be about 12 years old, and around whose waste [sic] I since measured the Creatures [sic] Skin" (Stedman 1988, p. 194).

As others have already keenly demonstrated, Stedman's anaconda or aboma cannot be extricated from the racialised and colonial logics in which the reptile is positioned, both pictorially and narratively (Lee 2002; Wood 2002; Bohls 2014). The engraving details Stedman, turned away from the viewer, pointing at the giant snake, which is subjected to dissection by David and hoisted up the tree by two other enslaved individuals. The gun responsible for dispatching the snake rests against the tree. Emily Senior (2010) aligns this Blake engraving with several others that punctuate *Narrative* to connect Stedman's treatment of violence against animals with violence against black and racialised subjects. She writes, "The flayed body of the snake is a troubling sight, and the nakedness of the black man carrying out the skinning serves only to enlarge the alarming nakedness of the writhing, skinless creature" (Senior 2010, p. 47). Indeed, with Blake's political abolitionism, his engraving of the aboma magnifies the horrors of slavery and the literal tortures enacted by Dutch and English forces against black and indigenous bodies, which are commemorated by his other engravings in *Narrative* that document lynchings, whippings, flayings, or what Mario Klarer (2005) has called "humanitarian pornography". The skinning of the aboma snake engraving highlights these human-animal elisions through colouration: in the facsimile version, the black individuals are shaded in identical ways to both the aboma's dorsal scales and the tree on which it is hanged. Stedman comments on this implicit connection, "I acknowledge [it] had a terrible appearance, viz, to See a Man Stark naked, black and bloody, clung with Arms and legs around the Slimy and yet living Monster" (p. 194). As Stedman's *Narrative* reminds us, black and enslaved complicity with these undertakings are requisites. There is then a colourised and racialised eco-logic that saturates the engraving, corresponding with Claire Jean Kim's (2015) assessment that the animal encodes struggles over race, species, and nature.[6] The texture of Blake's imagery then triangulates and overlays reptile, arboreal, and slave identities. The mass-reproduced

---

[4] As Richard and Sally Price, editors of the most complete edition of Stedman's unabridged manuscript, contend, Blake's rendering of Stedman's anaconda gave birth to a similar serpent that would appear in *America: A Prophecy* (1793). See the Price's Introduction, Stedman, 32.

[5] See "Stedman's Myrmecology: Decolonizing Analogy in Suriname" (Chow forthcoming).

[6] Kim focuses on the twentieth-century US in particular, but her claims have value for thinking about the use of the animal across global and historical spheres.

(and thus, more widely available) black and white facsimiles make this colourised logic even more palpable. To be clear, I am not suggesting that enslaved subjects are reptilian; I am instead drawing our attention to the ways in which Stedman's visual and narrative rhetoric interweaves black and serpentine identities and thus flays them identically.

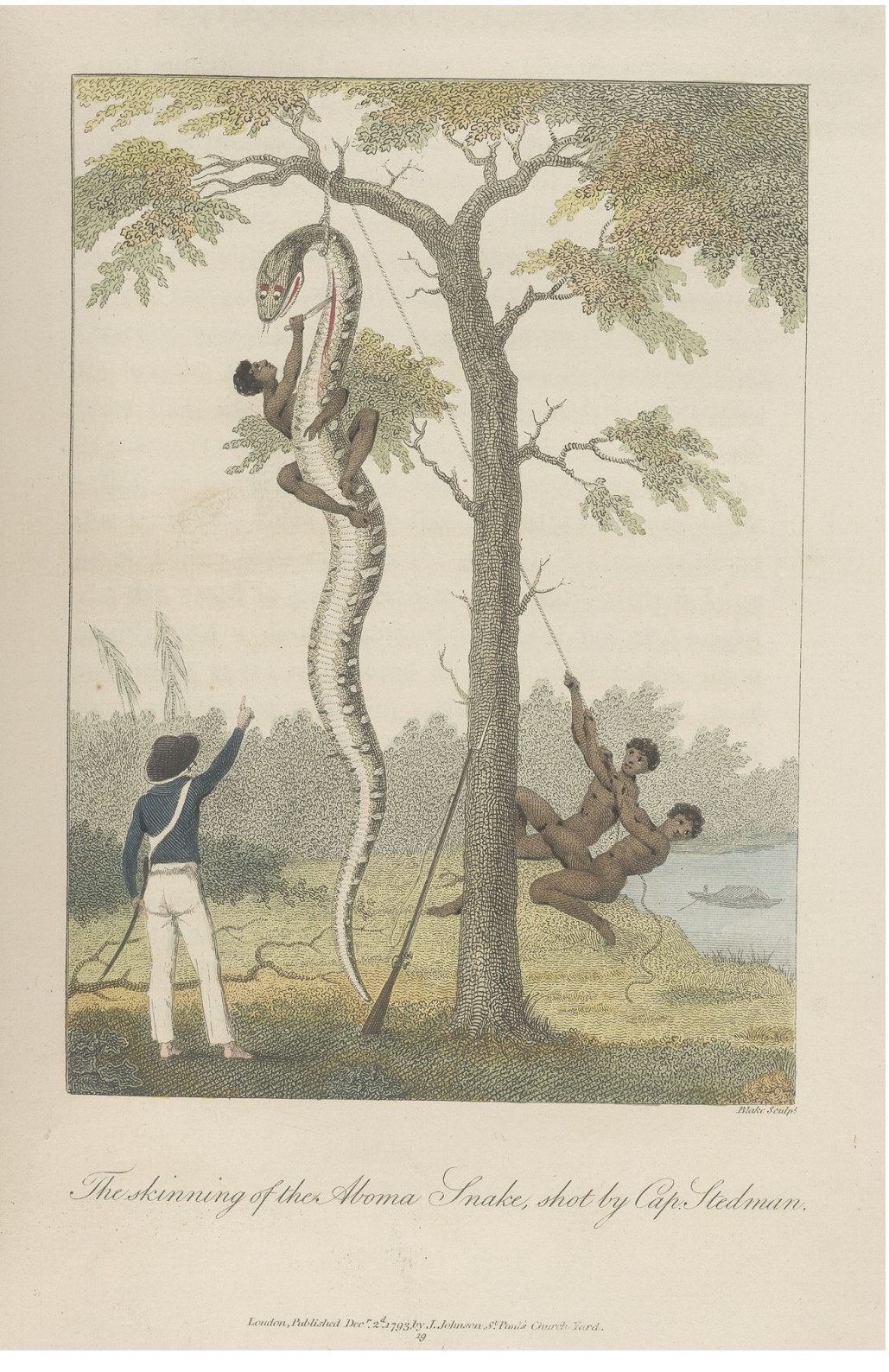

**Figure 1.** "The Skinning of the Aboma Snake, shot by Cap. Stedman" engraved by William Blake (1796). Public Domain.

### 3. Snake Lovers

Like the substitutive logics that inform our reading of Blake's engraving, Lewis's "The Anaconda" similarly relays the anaconda as a proxy by which to read multispecies relations and human-reptilian interchangeability. Because Blake's engravings of Stedman circulated so widely among reading publics, it is difficult to believe that Lewis was unfamiliar with the etchings or narrative (especially given his role as British attaché to the Hague in 1794 and his family's plantation-derived wealth in Jamaica), which unquestionably informs his incorporation of the anaconda. Greta LaFleur (2018) locates eighteenth-century natural history as germane to the emergence of sexuality studies, and while LaFleur's focus is primarily on the natural history of *homo sapiens* in early America, the attention to the fact that "eighteenth-century [natural history, in particular] texts imply, sexuality is not strictly a human experience but rather the effect of a combination of environmental factors that influence and at times find expression in human behavior", is worth considering here (p. 60).[7] In borrowing from East Indian and West Indian natural histories, Lewis's short story further situates LaFleur's claim that sexuality is not strictly a human experience; it is informed by its environmental placements and geographies, and, undoubtedly, its nonhuman interlocutors. While "The Anaconda" is replete with racist, racialistic, and colonialist vitriol, the titular anaconda is not an *exclusive* proxy for brown servant bodies. In perhaps its most evocative realisation, Lewis's narrative also relays the anaconda as a semaphore of a white, colonising body. The anaconda thus becomes an ironic narrative metonym for understanding the colonial contact zone of Ceylon, thus positioning a fantastical serpent as the central hinge upon which gender, race, and sexuality hinge in the East Indies.

In the short story's external-most framework, two mawkish, English busybodies hawk rumours miscommunicated by Everard's "coffee-coloured barbarian" (their phrase), who has borne witness to the anaconda incident (p. 5). The two collude to disengage Everard and his betrothed, Jessy, so as to preserve her virtue, chastity, and unblemished name. The scandalous knowledge they promote reveals that Everard has reaped mass wealth in tragic Shakespearean fashion. They claim that as a lowly secretary, he has ingratiated himself to a powerful wealthy English couple in Ceylon, cuckolding the husband and coercing the wife into poisoning her spouse so as to inherit the estate himself. The rumours further suggest that killing the husband was insufficient; Everard brutally murders the wife "in such a shocking manner too! First to shoot her from behind a hedge, and when he found the poor creature was only wounded, to have the heart to run up to her, and actually beat her brains out with a club!" (p. 5). In truth, the narrative later reveals that the anaconda causes the death of the husband, and by extension, the wife, and the rumours seem to capture the behaviours of the anaconda in Everard. By the busybodies' handling, the anaconda's actions are transformed and embodied by Everard, which demonstrates an ineluctable human-reptile elision. The external framework is then committed to upholding white heteronormative marriage, and while such a feature remains at the short story's end, the anaconda proves to complicate and tangle the apparent heteronormative longings—especially in the messy confusions surrounding agency, action, and embodiment.

The first illustration of the anaconda—a pseudo-ethology that mirrors Edwin's natural history—positions her, the pronoun by which she is identified, as rearranging notions of human and animal hierarchies. Zadi confirms,

> "When the Anaconda has once chosen a group of trees for her abode, and is seen to sport among their branches, in the manner in which we saw her amusing herself, she will remain there for whole days and weeks watching patiently for her prey, till every chance of success fails her, and absolute famine compels her to emigrate: But her capacity of existing without food is almost inconceivable, and till she removes of her own free will, no human power is able to drive her from her retreat." (p. 15)

---

7　Theresa M. Kelley (2012) likewise points to these blendings of human and plant-nonhuman expressions.

The determination and perspicacity by which Zadi characterises the anaconda imbues her nonhuman state with superhuman qualities. Zadi's invocation of the anaconda's "own free will" abuts larger late-eighteenth-century moral philosophical treatises on free will and moral considerability, promoted by Immanuel Kant and his philosophising contemporaries.[8] While Kant advocated against animal cruelty, his larger invocation of personhood clearly delineated human from nonhuman in what Erika Cudworth (2008) has called anthroparchy—an ethical and political allegiance to anthropocentrism that upholds human supremacy. In *Lectures on Anthropology* (1798) Kant observes, "The fact that the human being can have the representation 'I' raises him infinitely above all the other beings on earth. By this he is a person . . . that is, a being altogether different in rank and dignity from things, such as irrational animals, with which one may deal and dispose at one's discretion" (Kant 2013, p. 239). Lewis's anaconda is dispatched at the short story's end, which would seem to define her as befitting the genre of an irrational animal. Ironically though, the anaconda's presence and its accompanying sublimity prove to upheave human rationality: "But almost deprived of all the power of thought by their terrors, no one was able to point out any means for attacking her [the anaconda] with success" (p. 19). Still, the anaconda's ability to dispose of other animal life—a dog and a bull—in addition to killing Zadi and becoming directly responsible for, though not enacting, the deaths of Everard's patron, Seafield and his wife Louisa, demonstrates that the snake renders other forms of human and nonhuman life equally disposable. Lewis's anaconda thus becomes prepossessed with opportunities for rearranging systems of human and nonhuman relationality—and microcosmic biopolitical regimes that bifurcate the living and the dead—in striking, uncomfortable ways.

Everard's pseudo-natural history, which gives birth to his account of the anaconda, simultaneously enmeshes Zadi, the Ceylonese servant, in a nexus of intimate racialised, East Indian, and reptile affinities. As Julia Wright (2007) notes in *Ireland, India and Nationalism in Nineteenth-Century Literature*, "The Anaconda" typifies the colonial/oriental Gothic, which caricaturises bodies of colour and vilifies "oriental" locations as dangers to the "morality as well as the health of English national subjects, and to establish English virtue through the proper terror at oriental effects" (pp. 135–36). Part of these corrective, jingoistic effects derive from Zadi's pairing with the anaconda, which blurs human-serpent becomings. In other words, the narrative portrays Zadi as becoming an anaconda in East Indian human form. Such laminating logic continues the short story's racialistic vibes, which amplify long held natural history trajectories by which people of colour become affiliated with inferiorised animals in order to legitimate and herald white supremacy. However, unlike Donna Haraway's (2008) invocation of "becoming with", Zadi's reptilian proclivities are not framed as utopian or beneficent. In a gesture of self-sacrifice to save the man he has been forced to serve, Zadi "was to conceal his whole person from head to foot under a covering of boughs and cocoa-leaves resembling as much as possible the broken branches, with which the snake's gambols or indignation had stewed the hill all around her" (p. 25). The goal is to retrieve a letter from Seafield who finds himself immured in a pavilion to which the anaconda plays a hyper-vigilant gaoler. Everard watches with anticipation as "Zadi by a thousand serpentine movements reached the wall of the pavilion [ . . . ] The Indian's eye was fixed immoveably [sic] upon the snake, and followed all her twistings and windings with incessant application" (p. 26). Zadi is able to abscond with the letter because he physically embodies and replicates sinuous mobilities. He must abjure his humanness. Zadi's serpentine mimicry, Everard reasons, works inasmuch as it does not attract the anaconda's attention. By bedizening himself as serpentine bough, Zadi forges his connection with the anaconda—an ophidic detente of sorts, in that, by assuming a serpentine form, Zadi is neither under threat from nor threatening to the anaconda.

Because Zadi is disguised as vegetal life that moves in serpentine fashion, the anaconda and Zadi become narratively undifferentiated. They begin, in other words, to share an

---

8  See Christine Korsgaard (2018) for an extended discussion of eighteenth-century philosophies of rationality and their reflection or rejection of moral considerability in nonhuman animals.

ouroboros connection—a doubledness that becomes self-consuming. Everard recounts, "It required no less caution and dexterity to retire, than to approach; and never did I offer up more fervent vows, than at the moment, when the animated thicket began to set itself again in motion. Slower than the hour hand of a dial, now moving forwards, now backwards, now right, now left, it stole itself down the hill. Still it went on . . . and on . . . and lower . . . and lower" (p. 26). In what can be described as nothing less than thrusting movements, the "it" to which Everard refers identifies Zadi but could very well also correspond with the description of the anaconda. Inchworming his way thusly, Zadi becomes removed from his personhood—the pronoun "it" signifies this—and relegated to an "animated thicket". "Thick", from which thicket derives, as the *Oxford English Dictionary* reasons, becomes a descriptor of action, meaning quick or frequent, and figuratively engenders something gross, excessively disagreeable, or indecent (Thick 2020). The use of "thicket" here doubles down on Zadi's animalised rendering, which pushes the boundaries of decorum precisely because of the lewd anaconda performance that he enacts. However, in the pairing of the anaconda and animated thicket, the anaconda wins out; the snake ultimately detects Zadi's ersatz reptilian performance and "envelope[s] the unfortunate in her folds!" (p. 26). Again, Everard's suggestive language positions snake-likeness as the hinge upon which horrific eroticisms rest. The human-reptile elision here is not, then, just a recapitulation of anti-brown racism, but instead offers opportunities by which anacondas and animate thickets congress in graphic ways. As Everard beholds such an encounter, he notes, "a piercing shriek of horror burst from me! I felt all my blood congeal itself within my veins!" (p. 26).

Lewis's anaconda, though, is referred to as "she", a slippery pronoun usage, bequeathed by Zadi, that furthers the blurring of human and nonhuman lines, and makes the presumption of Everard as the anaconda even more provocative. For Wright, "[t]he anaconda itself is, on a number of levels, a typical orientalist symbol of the East: feminine, seductive, violent, difficult to see, prone to lethargy, and with voracious appetites, it at once terrifies and fascinates Everard" (p. 133). To the extent that Zadi becomes encoiled in the anaconda's "folds", so too does Everard become implicated in anaconda resonances. As with the tiger in Aphra Behn's *Oroonoko* (1688) and the Brobdingnagian monkey in Jonathan Swift's *Gulliver's* Travels (1724), the unstable and deliberately confusing ways by which eighteenth-century narratives pair biological sex, inconsistent pronoun use, and animal ineluctably manifests queer results.[9] These pronoun slippages demonstrate the instability of gendered performance and associations, which are further trivialised by the introduction of the animal, who is at once unknowable and yet documentable. Put another way, while human gender is itself slippery and unmoored, the introduction of the animal proves to exacerbate instabilities. The queerness apparent here stems not from the identification of the anaconda as a "she", but from the narrative's alignment of Everard, the masculine and imperialist hero and narrator, with the anaconda. The porosity of human and animal bodies signals these queer affiliations of gender.

The unfounded rumours circulated by the external framework's busybodies misinterpret the anaconda's actions as Everard's and thus reveal an unsettling lamination wherein moments of violence between a female snake and a male colonial secretary become presumptively shared. The two women begin their accusations by pinning Everard for the murder of a young woman he has promised to marry. The murder proves to be particularly unsavoury, "first to shoot her from behind a hedge, and when he found the poor creature was only wounded, to have the heart to run up to her, and actually bear her brains out" (p. 5). However, the women's accusations, in their unfounded state, misidentify the recipient of this violence. Whereas the women are certain that the violated individual is a fictitious woman named Nancy O'Connor, Everard's exonerative narrative reveals that the individual subjected to this violence is, in fact, the anaconda. Everard recounts, "I hastened to conclude this long and painful tragedy. I discharged my musket at the monster at a

---

9   See my essay, 'Prime Mates: The Simian, Maternity, and Abjection in Brobdingnag' (Chow 2020).

moderate distance. This time the ball struck her close by the eye. She felt herself wounded; her body swelled with spite and venom, and every stripe of her variegated skin shone with more brilliant and vivid colours" (p. 32). Violence between Everard and the anaconda becomes both co-created and narratively undifferentiated. He continues, "The report of my musket was the signal agreed upon to give notice to the expectant [Ceylonese] crowd that they might approach without danger. Every one now rushed towards the snake with loud shouting and clamours of joy. We all at once attacked her, and soon she expired under a thousand blows" (pp. 32–33). In truth, the actions of the Ceylonese and Everard are captured by the rumour mill: Everard stands as the imperial champion who has slain the monstrous anaconda with the assistance of the locals.

The rumours though position the fictional Nancy O'Connor as the anaconda, and in this way, the anaconda becomes further situated as Everard's surrogate lover. The women describe Nancy thusly: "Nancy was either the daughter, or the wife of a rich planter, with whom Everard lived as clerk [ ... ] this poor girl fell in love with Everard, and he on his side was wonderfully attentive to Nancy; for Mirza [Everard's Ceylonese attendant] says, that he passed whole days and nights watching her and ogling her, so that she actually could hardly stir without his knowing it" (pp. 5–6). The description reiterated by the two women is in fact corroborated by Everard. Yet it applies not to a Nancy O'Connor; instead, it characterises the female anaconda. Everard's "watching and ogling" are epitomised by his voyeurism of the anaconda. As a sublime idol that enraptures Everard's attention, the anaconda "united the most singular and brilliant beauty with every thing, that could impress the beholder with apprehension; and though while gazing upon it I felt that every limb shuttered involuntarily, I was still compelled to own, that never had I witnessed an exhibition more fascinating or more gratifying to the eye" (p. 17). The anaconda's sublimity becomes read as an erotic and romantic attachment, and thus, the rumours that seek to vilify Everard are not singularly concerns about domestic violence. They are frighteningly realised displays of human-snake eroticisms that when actualised must be jettisoned by and met with graphic violence, as Zadi's encounter realises. Anacondas, it seems, do not make for good lovers.

## 4. Serpentine Catalyst

Lewis's handling of the snake, though, is not distinct to his penning of "The Anaconda". Twelve years earlier, his *Monk*, which would shot-put him into celebrity, similarly invoked the snake in a tangential moment, which furthers the vermicular resonances that would later surface in "The Anaconda". However, with *The Monk*, the threat of a snakebite spurs a sensuous connection between Ambrosio—the titular monk—and Rosario/Matilda, the devil's apprentice. I see *The Monk*'s queer eruptions prompted by the snakebite to, as Tim Dean (2015) pointedly reminded us, recalibrate sex acts as central to understanding forms of queer relationality and theory.[10] That is, the introduction of a potential but false snake bite provides the opportunity for the corruption of priestly vows, oral sex, and arduous caresses to be mimetically performed by Rosario/Matilda, Ambrosio's proscribed lover. An alleged snakebite, put simply, becomes the impetus for pornographic eroticisms aplenty.

The monastery's garden (where else?) homes the passionate, lovelorn vows exchanged and broken by Ambrosio and Rosario/Matilda and becomes the Edenic locale for serpentine sensuality to burst forth. Forced with an ultimatum—either leave or "Stay, and you become to me [Ambrosio] the source of danger, of sufferings, of despair!"—Matilda (the former Rosario divested of the veil and male pronouns) pleads with Ambrosio for a vegetal token, which she will "hide [ ... ] in my bosom, and when I am dead, the Nuns shall find it withered upon my heart" (Lewis 2016, pp. 55–56). We see once again, as with Blake's engraving of the aboma and Zadi's bough-masquerade, the interwoven nature of plant life, animal life, and human passions is laid bare. Ambrosio acquiesces. "He approached the

---

10 While scholarship on *The Monk* often includes discussions of sexuality (viz. critiques of Catholicism or sensibility, or by way of Freudian or psychoanalysis), many frame these conversations outside the purview of queer theory. See, for example, David Jones (2014); Markman Ellis (2000); Anne Williams (1995); and David Punter (1996).

Bush, and stooped to pluck one of the Roses. Suddenly He uttered a piercing cry, started back hastily, and let the flower, which He already held, fall from his hand" (p. 56). As in Zadi's serpentine transformation in "The Anaconda", in the rosebush lays concealed the invocation of the snake. "'I have received my death!' He replied in a faint voice; 'Concealed among the Roses . . . A Serpent'" (p. 56). Ambrosio's hand engorges following the alleged snakebite and seeps with a "greenish hue" when subjected to the physician's lancet (p. 57). In editor Nick Groom's gloss, Ambrosio's articulation of the serpent harkens back to both Genesis 3:1 and Milton's retelling of pre- and postlapsarianism in *Paradise Lost* (1667). Although the serpent here is of course a red herring. Father Pablos, the physician, reveals, "From the sudden effects, I suspected that the Abbot was stung by a Cientipedoro", which is annotated by Lewis in the first and all subsequent editions as an insect "supposed to be a Native of Cuba, and to have been brought into Spain from that Island in the Vessel of Columbus"—yet another detail conjured by Lewis's fictional imagining (p. 57). The inclusion of the snake cum insect, a fictionalised West Indian invasive species no less, furthers the imperial matrix similarly found in "The Anaconda". The insect, believed to be a serpent, bites back.

The false snake's poison injected and seeping from Ambrosio's hand—it is later described as an 'orifice'—becomes the catalyst by which the venerated priest is overcome by nocturnal emissions, which are later realised through intercourse with Matilda (p. 57). From the poison-induced hallucinations, we learn:

> "But the dreams of the former night were repeated, and his sensations of voluptuousness were yet more keen and exquisite. The same lust-exciting visions floated before his eyes: Matilda, in all the pomp of beauty, warm, tender, and luxurious, clasped him to her bosom, and lavished upon him the most ardent caresses. He returned them as eagerly, and already was on the point of satisfying his desires, when the faithless form disappeared, and left him to all the horrors of shame and disappointment." (p. 66)

The shame and disappointment rendered here correspond with a variety of readings. In one, the shame and disappointment stem from the erosion of his priestly vow of celibacy, which the dream corroborates. In another, the shame and disappointment indicate the material post-ejaculatory reality that accompanies a powerful sex dream like this; shame and disappointment engender the enlightened, refractory state. In yet another, the shame and disappointment refer to two distinct entities: the shame that accompanies a priest's sexual relations and the disappointment that accompanies a dream such as this, without the promise of climax.

While in the former passage, Matilda's "most ardent caresses" remain imagined, in order for the envenomation to be eradicated, erotic contact becomes requisite. Father Pablos proves incapable of extracting the poison, and so, Matilda assumes the task as hers to conquer:

> "The Physician gave you over, declaring himself ignorant how to extract the venom: I knew but of one means, and hesitate not a moment to employ it. I was left alone with you; You slept; I loosened the bandage from your hand; I kissed the wound, and drew out the poison with my lips. The effect has been more sudden than I expected. I feel death at my heart; Yet an hour, and I shall be in a better world." (p. 69)

Ambrosio recognises Matilda's sickliness, which, she confesses, results from her own envenomated state: "Yes, Father; I am poisoned; But know, that the poison once circulated in your veins" (p. 69). As Lewis's narrative stylings are fain to do, Matilda's pronouncement of a share poisoned state and oral extraction of the green-pussed hand borders on the sexually explicit. Coleridge's (1797) critical review of the novel critiques the pseudo-pornographies that Lewis animates:

> "The temptations of Ambrosio are described with a libidinous minuteness, which, we sincerely hope, will receive its best and only adequate censure from the

offended conscience of the author himself. The shameless harlotry of Matilda, and the trembling innocence of Antonia, are seized with equal avidity, as vehicles of the most voluptuous images; and though the tale is indeed a tale of horror, yet the most painful impression which the work left on our minds was that of great acquirements and splendid genius employed to furnish a *mormo* for children, a poison for youth, and a provocative for the debauchee." (Coleridge 1797)

It's hard not to read the invocation of the *mormo* (a folkloric spirit used to discipline children by female caretakers), the poisoning of youth, and the debauchee's provocation as directly in conversation with this false-snake moment. In Ambrosio's hand, and then Matilda's body, course the poisons of passion that serve as cautionary tales for those yet unintroduced to the temptations of the flesh.

The eroticism indeed reaches pornographic levels when immediately before the disclosure of a shared intravenous poison, Matilda serenades Ambrosio accompanied by the harp. "The Songstress sat at a little distance from his Bed. The attitude in which She bent over her harp, was easy and graceful: Her Cowl had fallen backwarder than usual: Two coral lips were visible, ripe, fresh, and melting, and a Chin in whose dimples seemed to lurk a thousand Cupids" (p. 61). The vaginal imagery evoked by this description carries through to the curative kisses that Matilda provides in seeking to remedy Ambrosio's "orifice" that dribbles with a green venom. In her kiss lies the ardent caresses that have so fuelled Ambrosio's poison-induced dreams. The kisses, which I read as a surrogate fellatio, simultaneously enable the poison to flow between them as a shared bodily fluid that is exchanged from false snake to abbot to devil in disguise. As with Lewis's later "The Anaconda", the catalyst of the snakes provides opportunities by which to imagine the conjugality of enmeshed bodies as they sinuously form together; Lewis seems especially invested in the vaginal imagery that the anaconda's folds and Matilda's lips reproduce. "How dangerous", the narration reads, "was the presence of this seducing object" (p. 61). In the chapter's closing, it should come as no surprise that the two fully commit to sexual relations in ways that further the exchanges of venom that now infuse both bodies. Such a poisoned blending problematises heterosexual intimacy as unitive and reproductive. In sex between the devil and a priest, poison is mutually exchanged. "His kisses vied with Matilda's in warmth and passion. He clasped her hand rapturously in his arms; He forgot his vows, his sanctity, and his fame: He remembered nothing but the pleasure and the opportunity" (p. 71). The tangled tongues and enfolded hands reveal that the eroticism realised here spawns from the false snakebite.

## 5. Reptile Soul

Coleridge's snake imaginary, in Lewis-like fashion, draws the human-reptile connection along the axes of desirousness, queer intimacies, and supernatural phenomena. Matilda and Ambrosio's sensuous serpentine catalyst becomes the conduit to both the conferment of heteronormative intimacies—between Matilda as woman in male drag and Ambrosio, virile abbot—and of queer ones—Matilda's genderbending performance and alias as Rosario can reject the assumption that the intimacies with Ambrosio are anything *but* heteronormative.[11] Coleridge's *Christabel* pointedly aligns the snake alongside *female intimacies*. Gendered and queer readings of *Christabel* flourish, but in ways that I find do disservice to the enmeshment of human and animal bodies as vehicles for queer possibility that is so apparent to me (Gilbert and Gubar 2000; Taylor 2002; Baker 2010; Roulston 2019). I attempt to rectify that here with my own suggestive reading. While Ula Lukszo Klein (2019) and Lisa Jean Moore (2011) prefer the nomenclature of sapphic or lesbian, respectively, my use of female intimacies invokes Elizabeth Susan Wahl's (1999) earlier conception of female-female homoeroticisms throughout the Enlightenment, and Kristina Straub's (2009) more capacious use of intimacy in formations of gender and sexuality throughout the eighteenth century. I prefer the terminology of female intimacies and queer to consider mul-

---

11 For a trans-reading of the Matilda/Rosario bait-and-switch see Nowell Marshall (2017).

tivocal and polymorphic expressions of desire that abut, reject, erode, or just plainly situate heteronormative strictures otherwise, and need not account for the sticky transhistorical ways in which non-normative sexualities are repeatedly defined by action—specifically who one fucks. The poem's female intimacies emerge when the titular Christabel becomes the first participant in and then prey to the sinuous femininity of Geraldine. The occult vision of Geraldine as a "bright green snake" fosters Christabel's own serpentine affinities, which concretises the previous night's shared, denuded eroticisms (ln. 451).

Not published until 1816, Coleridge scribed the first part of *Christabel* in 1797 (the same year in which his review excoriated Lewis's *Monk*) and was unable to complete the second part until 1800; it did not make the cut of *Lyrical Ballads* (1798) because of Wordsworth's editorial dominance. The preface to John Polidori's 'The Vampyre' (Polidori 1819) reports that even in Coleridge's original 1816 oration—a result of the Gothic storytelling tête-à-tête that emerged from the sojourn to Villa Diodati upon Lake Geneva, and consequently birthed *Frankenstein* (1818) and 'The Vampyre'—the poetic snake's eyes inspired "cold drops of perspiration" in P.B. Shelley because of the imagined affinity with a familiar woman's breasts.

> It appears that one evening Lord B., Mr. P. B. Shelley, two ladies [Mary Shelley and Claire Clermont], and the gentleman before alluded to, after having perused a German work called *Phantasmagoria*, began relating ghost stories; his Lordship [Coleridge] had recited the beginning of *Christabel*, then unpublished, and the whole took so strong a hold of Mr. Shelley's mind that he suddenly started up and ran out of the room. The physician [John Polidori] and Lord Byron followed, and discovered him leaning against a mantelpiece, with cold drops of perspiration trickling down his face. After giving him something to refresh him, and upon enquiring into the cause of his alarm, his wild imagination had pictured to him the bosom of one of the ladies with eyes (which was reported of a lady in the neighbourhood where he lived), and he was obliged to leave the room to destroy the impression. (Polidori 1819)

The wild imagination that forces Shelley to abandon the sitting room stems from the Coleridge's (2020) repeated serpentine imagery and its feminine corollaries. In the neighbourhood woman's breasts lie the piercing—nearly Basilisk-like—glare of the snake. In truth, Christabel is likewise subjected to Geraldine's serpentine sensuousness.

The Gothic poem opens with ingenue Christabel, who absconds from her castle's comforts to materialise a fantasy held dormant in her dreams:

> "What makes her in the wood so late,
>
> A furlong from the castle gate?
>
> She had dreams all yesternight
>
> Of her own betrothèd knight;
>
> And she in the midnight wood will pray
>
> For the weal of her lover that's far away." (lns. 25–30)

Instead of her imagined knight, Christabel encounters the "Beautiful exceedingly!" and noble Geraldine, who has been abandoned in the forest and bonded to a tree under the threat of rape by five "warriors" (lns. 68, 81). In an ironic twist, Christabel becomes the knight she previously sought and Geraldine becomes the "maid forlorn" rescued by the virginal knight (ln. 82). These topsy-turvy inversions embellish the femme fatale trope that echoes so widely throughout the poem's two parts. "Femme fatales", Adriana Craciun (2002) writes, "in particular, with their inherent "doubleness" as both feminine and fatal, offer us an especially productive perspective on the development of sexual difference in the Romantic period" (p. 7). Craciun recalibrates understandings of femme fatales, not singularly as a pathology of male fears regarding female empowerment, but as an opportunity to visualise the modes of female sexuality, and female same-sex sexuality in

particular, that "account for the complexity of women's uses of seduction and violence in the Romantic period" (p. 16). Christabel and Geraldine cohabitate these same complex terrains.

The protectorate embodied by Christabel and offered to Geraldine becomes the central hinge by which to understand their female intimacy. A night of near-connubial bliss unfolds between the two as Christabel saves the weary and lost Geraldine by bringing the latter back to her room—"This night, to share your couch with me"—under the shroud of darkness and secrecy (ln. 122). Geraldine's frail state demands that Christabel overcompensate for the weary yet newly-acquired friend, and in so doing, the two re-enact a queer marriage act: crossing the threshold, one in the other's arms, into the consummate marital bed. The speaker reveals,

> "They crossed the moat, and Christabel
>
> Took the key that fitted well;
>
> A little door she opened straight, [ . . . ]
>
> The lady sank, belike through pain,
>
> And Christabel with might and main
>
> Lifted her up, a weary weight,
>
> Over the threshold of the gate:
>
> Then the lady rose again,
>
> And moved, as she were not in pain." (lns 123–33)

*Christabel*, in this way, plaits queer marital affinities that are resonant with, yet different from Jen Manion's recent *Female Husbands* (Manion 2020). "In their ability to flirt, charm, and attract female wives", Manion writes, "they [female husbands] threatened the stability of the institution of heterosexual marriage" (pp. 1–2).[12] In step, we see a proxy for this arrangement modelled by Christabel and Geraldine. One strand of the queer marriage unfurls from the surrogacy by which Geraldine becomes both the betrothed knight, sought by Christabel, and the maid forlorn. Positionalities become queerly swapped in that, like Geraldine, Christabel similarly becomes framed by the knight-maid dualism. The carrying of the lover over the "threshold of the gate" attaches yet another strand.

The final strand of the plait emerges from Christabel's invocation of her deceased mother whose final wish is to "hear the castle-bell / Strike twelve upon my wedding-day" (lns. 200–201). The poem opens, though, with a similar invocation of time "Tis the middle of the night by the castle clock, / And the owls have awakened the crowing cock" (lns 1–2), which is further relayed by the "toothless mastiff bitch" who operates as an animalised analog clock. "From her kennel beneath the rock / She maketh answer to the clock, / Four for the quarters, and twelve for the hour; / Ever and any by shine and shower, / Sixteen short howls, not over loud; / Some say she sees my lady's shroud" (lns 8–13). The sixteen short howls corroborate that it is in fact the middle of the night (exactly 12:15 a.m.), which summons the apparition of Christabel's mother. *Christabel* commences with the bewitching hour. The mother's dying wishes come to fruition, the reader learns, but through the union of Geraldine and Christabel. Geraldine likewise confirms the mother's surveillance of the queer connubial bliss when under her breath she expels the encroaching spirit whose presence again materialises: "'Off, wandering mother! Peak and pine! / I have power to bid thee flee.' [ . . . ] 'Off, woman, off! This hour is mine / Though thou her guardian spirit be, / Off, woman, off! 'tis given to me'" (lns 105–13). The maternal ghost's omniscience unveils Geraldine's sinister motivations, and yet, in the queer marriage performed, enacted, and consummated, this ghost likewise becomes an ethereal and heteronormative intervention that seeks to censure female intimacy. Indeed, as Robbie B.H. Goh (2003) observes, Geraldine desires to not only eschew the mother, but replace her altogether: "Can this be she, / The lady, who knelt at the old oak tree? / And

---

12    Manion's conception of the female husband refers to women who assume 'the character of a man' and thus crossdress or emerge as proto-trans figures (p. 2). This is not the case for *Christabel*.

lo! the worker of these harms, / That holds the maiden in her arms / Seems to slumber still and mild, / As a mother with her child" (lns. 295–300). The layers of female intimacy unfold in these crosshatched surrogacies.

Whereas effusions of sensuality emerge from the poem's start, the serpentine sensuousness surfaces only after the queer marriage and sex acts; Geraldine's reptile form appears and thus extends plural forms of queer intimacy that at once foreclose connubial consummation, but spur Christabel's own reptilian transformation. Christabel's father, Sir Leoline, upon hearing of Geraldine's tribulations demands justice, by way of "tourney court—that there and then / I may dislodge their reptile souls / From the bodies and forms of men!" (lns. 441–43). Geraldine recoils at the invocation of "reptile souls"—she was "ruthlessly seized"—which appears to Leoline as a gesture of overwhelming gratitude. However, Leoline's clairvoyant invocation of a reptile soul unwittingly unearths Geraldine's reptilian form, and the seizure she bodies forth becomes not a gesture of gratitude, but a woeful recognition of her uncloseting. Put another way, Geraldine shutters that her snakeskin has been outed. Leoline's bard likewise pinpoints Geraldine's alleged ophidic state when he recounts his previous night's dream. Bard Bracy recounts:

"And in my dream methought I went

To search out what might there be found;

And what the sweet bird's trouble meant,

That thus lay fluttering on the ground.

I went and peered, and could descry

No cause for her distressful cry;

But yet for her dear lady's sake

I stooped, methought, the dove to take,

When lo! I saw a bright green snake

Coiled around its wings and neck.

Green as the herbs on which it couched,

Close by the dove's its head it crouched;

And with the dove it heaves and stirs,

Swelling its neck as she swelled hers!

I woke; it was the midnight hour,

The clock was echoing in the tower;

But though my slumber was gone by,

This dream it would not pass away—

It seems to live upon my eye!" (lns 543–61)

In identifying Christabel as the dove, the bard's sonic narrative frames the transmogrifying nature of human and animal interchangeability. For bard Barcy, Geraldine's serpentine affinity is both a horrific dream and a horrific reality: Geraldine's embrace of the naive Christabel becomes a coiled suffocation that anticipates the engulfing of the latter in the former's body. As with "The Anaconda", the observation of serpentine appetites here rings with echoes of a sublime violation of the body: framing women, in particular, as victims of rumoured domestic violence by domineering paramours. Alongside the erotic snake catalysts in *The Monk*, the vision of Coleridge's snake's bright green colouration and swelling neck also evokes animalised renderings of graphic eroticisms. Geraldine as a snake sutures these modes together: her grasp of Christabel induces a frightening domestic violence that damages the connubial couch-sharing experienced the night before, and at the same time, becomes a colourful narration that illustrates a female intimacy between the two as an open secret shared with Leoline and the court. To see Geraldine in her serpentine form is then to reckon with levels of female intimacy that entangle the two women in forms of graphic sex that is otherwise read as domestically violent. Even so, not all graphic sex

is violent or non-consensual, and this is the subversive polemic I read into Geraldine and Christabel's connected serpentine states. In other words, Christabel provides opportunities for envisaging diverse representations of female sexual agency that are consensual, rough, graphic, and illegible to others.[13]

The poem's quick movement to a close seemingly demands the ejection of Geraldine from the court, not only to be reunited with her own father, but at the behest of Christabel, who now beholds her one-time lover in metamorphosed reptilian form with anguish and fear. Such a pat reading of the poem's conclusion suggests that the eschewal of Geraldine coincides with the eschewal of the snake and queer reptilian intimacy; their tethered nature can be jettisoned *en masse*. I prefer a different reading: the poem's ending positions Christabel, once more, as Geraldine's saviour, so as to further secure the female intimacy previously enjoyed, as well as in recognition of her own reptilian transformations. Ignorant to Christabel's and Bracy's conjoined vision of Geraldine as snake, Leoline's chivalry seeks to eradicate the snake as a long-held symbol of vice and sin—Leoline becomes a sort of St. Patrick incarnate. He orates to Geraldine, "Sweet maid, Lord Roland's beauteous dove, / With arms more strong than harp or song, / Thy sire and I will crush the snake!" (lns. 571–73). Leoline misapprehends the bard's words and instead imagines the snake as a threat to Geraldine's chastity. He fundamentally misunderstands that in the promise to crush the snake, with Geraldine's father no less, that they intend to crush Geraldine. Upon making this promise, Christabel witnesses Geraldine transform:

"A snake's small eye blinks dull and shy;

And the lady's eyes they shrunk in her head,

Each shrunk up to a serpent's eye

And with somewhat of malice, and more of dread,

At Christabel she looked askance!" (lns 585–90)

Again, the invocation of serpentine affiliation births a fully realised snake in female form—a trope that similarly takes root in Keats's *Lamia*. Coleridge and Keats thus toggle back and forth imagining the baleful accounts of snake women. Whereas Lamia's divestment of her human form precedes the consummation of her marriage, Geraldine's unveiling proceeds from the consummation of the relationship on the connubial couch. Christabel pleads with her father, "By my mother's soul do I entreat / That thou this woman send away!" (lns. 618–19). Although Christabel's mother remains antagonistically opposed to Geraldine throughout the poem (demonstrating perhaps their fungibility), the plea to send Geraldine away is not predicated on the fear of Geraldine as snake. Put simply, Christabel may have fears but ophidiophobia is not one of them. Christabel's fear realised is her father's promise to "crush" the snake, which would simultaneously serve to crush her lover. Christabel cannot allow the threat of violence, wielded again by domineering men, against Geraldine—an echo of their initial encounter. Christabel's pleas safeguard Geraldine from violation or threat as well as sanctify the eroticisms they share. Crushing the snake thus becomes a pseudo-euphemism that disavows female intimacy and subjects female intimate partners to physical violence by unwelcomed male partners.

Such a protection becomes even more palpably felt: throughout the poem, Christabel has audibly morphed into snake form. Reptilian embodiments, in other words, become contagious in *Christabel*. Following her father's pronouncement, and witnessing Geraldine in her reptilian body, "Christabel in dizzy trance / Stumbling on the unsteady ground / Shuttered aloud, with a hissing sound" (lns. 591–93). This is not Christabel's first hissing sound; it likewise accompanies Christabel's previous vision of Geraldine as a snake in her mind's eye: "Again she saw that bosom old / Again she felt that bosom cold / And drew in her breath with a hissing sound: / Whereat the Knight turned wildly round, / And nothing saw, but his own sweet maid / With eyes upraised, as one that prayed"

---

[13]   I am gesturing towards the possibility of reading kink—and potentially BDSM subcultures—in Geraldine and Christabel's relationship. Samuel Rowe's (2016) discussion of meter and prosody in the poem attends to this subliminal current more explicitly.

(lns. 457–62). Christabel's onomatopoeic hisses attract the attention of her father, an aural indicator that notices not only serpentine sensuousness but also a mark of female intimate relations. To hiss, as Christabel does, is to incriminate herself, which is captured by a recognition that Christabel makes the morning after, "'Sure I have sinn'd!' said Christabel, / 'Now heaven be praised if all be well!'" (lns. 381–382). The sibilant sounds of Christabel's admission extend the aural quality of the hiss and invite readers to embrace the erotic soundtrack of Christabel's serpentine transformation—a transformation that aspires to embrace Geraldine in both human and reptile embodiments. In "The Anaconda", the sight of the indistinguishable serpent becomes the impetus for queer entanglements; in *The Monk*, the invocation of a snake's touch fosters erotic impulses; and in *Christabel*, it is both the vision of the snake and its accompanying aurality that make possible the female intimacy of human-reptile unions. By Coleridge's handling, to relish the beauties of female intimacy is to likewise relish the pleasures that anguilliform avails.

### 6. Serpentine "Colours of the Rainbow"

The Gothic snake is a material and lexical semiotic that demands attention. Gothic multispecies encounters draw our attention to the agentive, cognitive, embodied, and affective possibilities the emerge from human and nonhuman interactions. As animal studies scholar Lori Gruen notes, "Our relationships with humans and animal others co-constitute who we are and how we configure our identities and agency, even our thoughts and desires" (Gruen 2015, p. 63).[14] This essay has queried the types of thoughts and feelings that the recurrence of the snake inspires, which ultimately captivate Gothic authors and their audiences. The snake's material body becomes an emblem by which to wink-wink-nudge-nudge readers to recognise moments of queer and female intimate desires. By looking to a Gothic constellation of human-snake intimacies—whether literalised, imagined, or conjured—I have shown here that this multispecies connection is replete with a queer affinity that hinges upon forms of bodily penetration and plural eroticisms—realms of vulnerability where myriad queernesses blossom. When the Gothic mind's eye conjures the snake, we witness the power of human-reptile relations to upheave and subvert heteronormative notions of intimacy. Track the snake, follow its limbless movements, and witness its slippery mobility that portends new modes of relation. Beware of the snake, these narratives remind us, because under its iridescent shimmer lies a rainbow of erotic entanglements, perhaps best characterised by Lewis's Everard, who describes the anaconda thusly: "for with the animal's slightest movement all these points, and spots, and contrasts of variegated hues, melted together in the sun beams, and formed one universal blaze composed of all the colours of the rainbow" (p. 17).

**Funding:** This research received no external funding.

**Institutional Review Board Statement:** Not applicable.

**Informed Consent Statement:** Not applicable.

**Acknowledgments:** Thanks is owed to Ghislaine McDayter whose support and feedback have made this essay stronger.

**Conflicts of Interest:** The author declares no conflict of interest.

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
