# Peer review of "Snaking into the Gothic: Serpentine Sensuousness in Lewis and Coleridge"

_humanities, doi:10.3390/h10010052_

Round 1
Reviewer 1 Report
This essay intersects ecocriticism, empire, and queer studies to illuminate the eroticism of snakes and snake imagery in Romantic Gothic. Its argument is unique and provocative and the essay is well-researched and supported. One of the strongest contextual sections discusses Blake’s illustration of the Aboma Snake. The author clearly demonstrates that foreign snakes and reptiles were a part of the visual imagination as the British Empire saw global reach. The connection between depictions of animals and the position of the enslaved subject is relevant. The essay’s close readings of Gothic literature, bodies, race, and empire are engaging and persuasive. The most exciting section discusses Lewis’s short story “The Anaconda.” (Perhaps noting what scholarship has previously discussed the story would be helpful for the reader).
While the author provides such convincing close readings, the section on “Christabel” needs to engage more closely with previous scholarship on the poem, snakes, and femme fatales.
Does the essay need six subsections? Is section 6 needed?
If possible, the essay may benefit from illustrations of snakes from the literary examples the author discusses (or eighteenth-century illustrations of the snake and centipede).
There are some errors (“P.S. Shelley”; several dangling modifiers) that will need addressed. Some words include British spelling (such as “realise”) but this is not consistent in the manuscript. The “S” alliterations throughout will need editing (starting with the title).
Author Response
Dear Reviewer & Editor,
I want to first express my gratitude for your time in reading and considering my piece for publication in Humanities, especially with such a prompt turnaround. I have attempted to respond to the constructive criticisms provided in the reviews in order to streamline the piece and clarify any concerns. In addition to working on polishing the prose, I have revised the piece in three major areas:
Situating ‘The Anaconda’
Because Lewis’s short story is not anthologized and has received minimal criticism and attention, I have found it challenging to ensure that I am framing my discussion with extant scholarship. Over the revision period, I have been able to find at least one relatively recent work that directly engages the short story, and I have placed myself in conversation with that scholar, Julia Wright, more pointedly in this draft. Wright is doing something different than I am, but I find that her reading of the short story as an emblem of cross-cultural imperialism is in mode with what I am suggesting about the short story’s imperial framing of race, gender, and sexuality.
Situating Christabel
The first reviewer encouraged me to consider how I might reframe my reading of Coleridge’s poem around the femme fatale. I have taken this recommendation to heart and attempted to situate my reading of Geraldine and Christabel as equal femme fatales in ways that further the queer reading I offer here. In addition, I have also better located Christabel in its Romantic contexts, especially in its original oratory form. While much has been said about Christabel, I find that my citational model shown here demonstrates those with whom I see myself in conversation--so that my reading doesn’t necessarily stand alone, but certainly charts more nuance territory.
Conclusion Revisions
The first and the third reviewer offered mixed reviews of the conclusion. I will note that while I feel strongly about the pedagogical implications of my conclusion (because my teaching informs my research and vice versa), I have, in this draft, omitted that discussion. One of the reasons for removing this section was because of the article’s already robust length. That said, should the editors and reviewers see this as an important aspect of the longer piece and offer an important connection to the other pieces in this special issue then I would welcome the opportunity to reincorporate it in the final draft.
I believe my revisions attached here effectively respond to the reviewers, and the piece is better for it. I welcome feedback from the editors as this article continues to move towards publication.
Reviewer 2 Report
I am admittedly new to these contemporary theories, but I found the paper to be an extremely original and insightful exemplar of their application. Probably because I have done very little with queer theory, I found that aspect of the essay somewhat confusing at first, though it became clearer in the course of the argument. That is why I suggested clarifying that particular aspect of the methodology. Then again, perhaps readers well versed in queer theory need no such clarification.
Author Response
I want to first express my gratitude for your time in reading and considering my piece for publication in Humanities, especially with such a prompt turnaround. I have attempted to respond to the constructive criticisms provided in the reviews in order to streamline the piece and clarify any concerns.
Regarding the variety of queer methodologies that I bring to bear on this project, I have attempted to clarify how each text offers a different queer approach. The first section approaches queer readings of gender, sexuality, and racial formations; the second examines the pornographic associations with snakes; and the last demonstrates the violent yet loving female intimacy shared by Christabel and Geraldine. The introduction attempts to preface these various interventions while allowing each section to grow in various directions. As the reviewer notes, I do believe that since the special issue is on queer approaches to eighteenth-century studies that the readers will potentially have a particular buy-in to this approach and be familiar with some of the methodologies that I offer.
Reviewer 3 Report
This is a thoroughly researched, very readerly, and original article that makes (as it promises) key interventions into Gothic, gender, and animal studies. It does by enriching the reader's literary and scientific understanding of the serpentine within a variety of eighteenth and nineteenth centuries. The article is persuasively and engagingly written and culminates in a fresh reading of Coleridge's 'Christabel' that offers a new way of (re-)thinking the poem's poised and unresolved ambiguities.A couple of stylistic comments.
Perhaps, best to avoid contractions (such as I've etc.). The phrase 'specifically who you fuck' has rhetorical force, but the editors of the journal may have a view about whether this is an appropriate register of language. The meaning would be retained if it read 'who your sexual partners are'. I enjoyed the reflective piece about pedagogy and zoo visits which acts as a coda and conclusion to the article. Although this might have served equally well as the article's opening gambit.
One comment, in particular, I would be grateful if you could pass on to the author is that to complete his / her thorough research into the serpentine, is that s/ he add a reference in the bite to Teddi Chichester (concerning Shelley's 'The Assassins' and 'On the Medusa') a reference to Jerome McGann's 'The Beauty of the Medusa: A Study in Romantic Iconography', Studies in Romanticism 11 (1972): 3-25.
Author Response
I want to first express my gratitude for your time in reading and considering my piece for publication in Humanities, especially with such a prompt turnaround. I have attempted to respond to the constructive criticisms provided in the reviews in order to streamline the piece and clarify any concerns, especially with regards to prose.
The citational recommendation was especially helpful and reminded me that I was familiar with the article (it had been recommended to me previously) and had forgotten to include the citation in the draft. I have corrected that here.
The first and the third reviewer offered mixed reviews of the conclusion. I will note that while I feel strongly about the pedagogical implications of my conclusion (because my teaching informs my research and vice versa), I have, in this draft, omitted that discussion. One of the reasons for removing this section was because of the article’s already robust length. That said, should the editors and reviewers see this as an important aspect of the longer piece and offer an important connection to the other pieces in this special issue then I would welcome the opportunity to reincorporate it in the final draft.
Reviewer 4 Report
Essay provides a novel investigation of the serpent figure in several important classical Gothic works. The readings of The Monk and Christabel make an original and beneficial addition to the scholarly work to date. The introduction of Lewis's short story is welcomed, as it is, in the author's own terms, currently understudied. The queering of the texts studied through a close attention to serpent/human interaction, entanglement, and induced "slippery mobility" convincingly illustrates the non or anti heteronormativity lurking below the surface of the gothic facade in these works.
I do not have any suggested improvements for the author, except a couple of minor typos:
P. 9 (top): Zadi's name is misspelled.
P. 13 (lines 11 & 12): "both" appears erroneously repeated.
Author Response
I want to first express my gratitude for your time in reading and considering my piece for publication in Humanities, especially with such a prompt turnaround. I have attempted to respond to the constructive criticisms provided in the reviews in order to streamline the piece and clarify any concerns, especially with regards to the prose.